# The Wheat Aleurone Layer: Optimisation of Its Benefits and Application to Bakery Products

**DOI:** 10.3390/foods11223552

**Published:** 2022-11-08

**Authors:** Lucie Lebert, François Buche, Arnaud Sorin, Thierry Aussenac

**Affiliations:** 1Foricher Les Moulins, 95400 Arnouville, France; 2Institut Polytechnique UniLaSalle, Université d’Artois, ULR 7519, 60026 Beauvais, France

**Keywords:** wheat aleurone, dietary fibre, extraction process, antioxidant, bread, arabinoxylans

## Abstract

The wheat aleurone layer is, according to millers, the main bran fraction. It is a source of nutritionally valuable compounds, such as dietary fibres, proteins, minerals and vitamins, that may exhibit health benefits. Despite these advantages, the aleurone layer is scarce on the market, probably due to issues related to its extraction. Many processes exist with some patents, but a choice must be made between the quality and quantity of the resulting product. Nonetheless, its potential has been studied mainly in bread and pasta. While the nutritional benefits of aleurone-rich flour addition to bread agree, opposite results have been obtained concerning its effects on end-product characteristics (namely loaf volume and sensory characteristics), thus ensuing different acceptability responses from consumers. However, the observed negative effects of aleurone-rich flour on bread dough could be reduced by subjecting it to pre- or post-extracting treatments meant to either reduce the particle size of the aleurone’s fibres or to change the conformation of its components.

## 1. Introduction

Wheat is indispensable in producing many staple foods around the world, including bread, biscuits, cakes and noodles. As epidemiological studies have demonstrated that an increase in the intake of whole grain products is related to a lower incidence of cardiovascular disease, obesity, diabetes and cancer, the composition of wheat makes it a valuable asset in the diet for the prevention of chronic diseases. Consequently, in relation to the growing number of metabolic diseases, nutritional guidelines worldwide advise an increase in the consumption of whole grain products, partly as they contain fibres that many consumers lack in their diet [1,2].

These health benefits are mostly due to the presence of micronutrients, dietary fibres (DF) and bioactive components, which are mainly located in the outer layers of the grain: the bran and the aleurone layer [1,3,4]. Many researchers and industries have aimed to extract, isolate and introduce these grain fractions in food products as ingredients for added nutritional value. However, it seems that the addition of bran or fibre to wheat-flour-based products changes not only the technological properties of the end-product but also its sensory acceptance by consumers [3,5]. Moreover, the main challenge of the extraction process is improving the nutritional properties of the ingredient without impairing its technological properties during breadmaking. For instance, soluble fibres, such as water-extractible arabinoxylans (WEAX), answer this problem. The use of fibres from wheat bran has also been deeply investigated [3,6,7,8].

Recently, researchers have focused on the wheat aleurone layer, considered by millers to be the main bran layer. As it contains the majority of the grain’s minerals and is also rich in protein, DF and bioactive components (mostly ferulic acid), the aleurone layer may be the source of many bran’s reported health benefits [4,9,10,11]. However, despite its known nutritional and health-prevention properties, the aleurone layer is scarce on the market, whether included in cereal food products or as an ingredient. This could be related to the challenges posed by its extraction. Although multiple processes have been patented, the end product is often obtained with either low purity or yield [10,12,13,14]. Moreover, it seems that its incorporation into bakery products yields contrasting results, as some negative technological effects could be observed, such as a reduction in loaf volume and increased crumb hardness [15,16,17]. These limiting technological aspects on final food products could be improved by making specific modifications to the aleurone’s components without losing any of the health benefits, prior to its incorporation into a food process, as is already carried out for wheat [18].

The objective of this review, therefore, is to provide readers with the elements of understanding to optimise the potential of the wheat aleurone layer, both from a nutritional and technological point of view. To do so, the wheat aleurone layer will first be described in terms of function and composition. Its potential nutritional and health benefits will then be presented based on clinical studies and in relation to the individual effects of its compounds. Aleurone’s potential as an ingredient will also be reviewed, starting with the processes to extract it and related issues. Then, the applications of cereal-based food products performed in the literature will be investigated, with a tentative explanation of the underlying mechanisms of the observed effects. Finally, the last section will focus on the existing processes meant to optimise the aleurone layer’s potential, both for its nutritional, health-related and technological benefits.

## 2. Aleurone Layer

### 2.1. Description: Histology and Functions

The aleurone layer is a tissue of wheat grain made of unicellular block-shaped cells (37–65 µm vs. 25–75 µm) [19]. Among the seven layers comprising the mature bran, the aleurone is the only one with the remaining living cells [20]. It is located between the endosperm and the nucellar epidermis (or hyaline layer), as shown in Figure 1 [10,21]. Although botanically part of the endosperm, it is considered by millers as a bran layer since it remains attached to the hyaline layer during milling [10,22]. It represents around 50% of wheat bran (or 75% *w*/*w* of its dry weight), making it the major bran layer [10,12]. Indeed, the aleurone layer is thick and can reach up to 65 µm [23]. It also corresponds to 5–8% (*w*/*w*) of the whole kernel [21].

Although the aleurone layer is singular in wheat, it can be found multi-layered in barley, rice and oats [10].

Multiple functions in the wheat grain are allotted to the aleurone layer, namely: accumulation and transport of nutrients for seed germination, decomposition of storage materials of the endosperm for embryo growth, and protection and maintenance of caryopsis activity [24]. More specifically, its major role is during germination, where it is involved in the synthesis and release of hydrolases in the endosperm, as induced by gibberellin [11,25]. These enzymes then break down starch polymers and proteins in starchy endosperm cells, which undergo programmed cell death [22,26]. To facilitate the transfer, the aleurone’s outer cell wall is degraded by endogenous hemicellulases, and only the inner resistant layer remains [25,27].

The aleurone layer is also involved in seed dormancy, induced by abscisic acid. At the same time, this hormone induces programmed cell death in endosperm cells [11]. However, the aleurone layer serves as grain storage for metabolites, minerals and amino acids [9,26,28]. It is equally involved in the regulation of water diffusion and distribution through its cell-walls [11,29].

Finally, in the crease region of the grain, some modified aleurone cells, called transfer cells, also participate in grain filling via solute uptake [30]. Due to their arabinoxylan’s higher degree of arabinose substitution and lower degree of feruloylation, the transfer cells show specific cell wall hydration and porosity properties that are compatible with water diffusivity and uptake for grain filling [31].

### 2.2. Composition

#### 2.2.1. Cell Wall

The aleurone cell-wall represents 35% (*v*/*v*) of the total cellular volume. It is bilayered, with a thicker outer layer (of 2 µm vs. 0.5 µm for the inner layer) and mainly composed of arabinoxylans (65%), β-glucans (29%) and phenolic acids [9], as presented in Table 1 and Table 2. This high proportion of dietary fibres (DF), especially pentosans (44% *w*/*w* of total grain [32,33]), makes the aleurone’s cell walls valuable from a technological and nutritional point of view, thanks to their gelling ability [6], and cancer-prevention properties [34,35].

Its major component, arabinoxylan (AX), is a polysaccharide made of a linear xylose chain with β-(1,4) linkages between its xylopyranosyl residues (Figure 2 [38]). These residues can either remain unsubstituted, mono-substituted at the third carbon position (C3), or with a di-substitution at both the C2 and C3 positions. Indeed, α-L-arabinofuranose residues can be esterified to the xylan backbone via α-(1,2) and α-(1,3) linkages but not in an even pattern. These arabinose residues can, in turn, be esterified by phenolic acids (mainly ferulic acid) on their primary alcohol function (O5) [9,10,39]. The feruloylation of AXs is progressive throughout grain filling [31] and takes place on average every 15 arabinofuranose residues [22]. In addition to arabinose substitution, some acetyl groups can be esterified to xylan instead [30]. The presence of these side groups confers technological value to aleurone’s AXs as they can interact with AX chains (dimerization) or other molecules, such as proteins or fibres, thus strengthening the gluten network [6].

With a low arabinose to xylose ratio (A:X = 0.41–0.47), aleurone cell wall AXs are considered poorly cross-linked [23]. AX solubility is thus influenced since the presence of arabinose residues causes the formation of long asymmetrical polysaccharides; hence, an unsubstituted region has a tendency to aggregate (stabilisation by hydrogen bonds) and become water insoluble [6,25,40]. This insolubility increases with the presence of FA, which is shown by a high FA monomer-to-arabinose ratio (7.2–7.4) [10,23]. Only a few water-extractable AXs (WEAX) can be found at the cell-wall surface (1.5 to 2.5%), probably related to incomplete cross-linking with other components [39]. In addition, compared with other bran layers, there is less cross-linking between polysaccharides and phenolic compounds in aleurone. This might be due to the role of aleurone in the enzymatic degradation of the stored compounds, which requires facilitated movement through the cell walls [41]. Although, most health benefits related to the presence of AXs in the diet stem from the addition of WEAX, the predominance of water-unextractable AXs (WUAX) in the aleurone cell-walls can also be valued. However, it is often associated with negative rheological traits in breadmaking [39].

The second main component in wheat aleurone cell walls, β-glucan (BG), consists of a linear chain of glucose residues joined by glycosidic linkages [9]. The majority of the chain (90%) is made of cellotriosyl (DP3) and cellotetraosyl (DP4) units bonded by β-(1,3) linkages. The remaining 10% refer to β-(1,4)-linked side chains [25]. According to Jamme et al. [42], BGs are not present in the junction zone, whereas they are abundant in the inner periclinal face. As presented in Table 1, there are more BGs in aleurone cell walls compared to bran, which can be related to the layer’s reported health benefits, such as the modulation of digestion [43].

Finally, phenolic acids are present in the aleurone cell walls (Table 2). They are mostly (95%) represented by ferulic acid (FA) in its trans form (90%) [44]. Their presence increases cell wall resistance by forming covalent cross-links (esterification) between AX chains through oxidative dimerization [25]. The most common dimerised forms are 5-5′-diferylate, 8-O-4′diferulate, 8-5′diferulate and 8-5′benzo diferulate [9,44]. Moreover, 92% of FA is found in bound form in the aleurone layer and accounts for 55 to 60% (*w*/*w*) of the whole FA grain concentration according to Barron et al. [23], but can reach up to 70% (*w*/*w*) as per Brouns et al. [10]. This is highly beneficial, as FA is renowned for its antioxidant properties, which increase the antioxidant capacity of the aleurone layer [45].

Other phenolic acids include para-coumaric acid (PCA), which constitutes 67% (*w*/*w*) of the whole grain concentration and about 10% (*w*/*w*) of the total phenolics in aleurone cells [23]. It is also 3 to 5 times more concentrated in this bran layer than in the others but is not found in the endosperm [46]. Mainly in bound form (63% *w*/*w*), PCA is substituted to AX on average once every 90 arabinofuranose residues [10,22]. Sinapic and vanillic acids can also be found, mostly in conjugated form, amounting to 69 and 67%, respectively [10,23]. Traces of free syringic acid and bound flavonoids (apigenin and lutolein) have also been reported in the cell walls of the aleurone layer [10,47,48]. These phenolic compounds contribute to the total antioxidant capacity of the aleurone layer.

**Table 2 foods-11-03552-t002:** Phenolic acids repartition in the wheat aleurone layer, bran and whole grain.

	Aleurone Layer * (mg/100 g dm)	Bran * (mg/100 g dm)	Whole Grain (mg/100 g dm)
Ferulic acid Total Monomer Dimer Trimer	628–817 [32,49] 798–814 [23] 31–107 [23,32,49] 2–15 [23,49,50]	500–1500 [3,49] - 101 [49] -	- 86–87 [23] 14–15 [23] 3–4 [23,49]
p-Coumaric acid	15–29 [23,49,50]	13–16 [10]	1–3 [10,23,49]
Sinapic acid	6–44 [10,23,46,49]	11–28 [10]	4–8 [10,23]
p-hydroxybenzoic acid	2.8 [10]	1.9–2.2 [10,49]	0.5 [10]
Vanillic acid	2 [10]	1.6–3.5 [10,49]	0.5–2.1 [10]
Syringic acid	9 [10]	3.5–5.7 [10,49]	1.3–1.8 [10]

* The aleurone layer and the bran amount respectively to 5–8% (*w*/*w*) and 5.5–9.5% (*w*/*w*) of the wheat grain [21]. All data were placed in the same unit to facilitate comparison.

Aleurone cell wall polysaccharides (AX and BG) are in different proportions depending on their position in the bilayered cell wall. Although the thinner inner layer seems to comprise more BGs, the thicker outer layer contains more AXs. FA residues in both layers appear unchanged, although there seem to be more phenolic acids in the anticlinal than periclinal walls [25,51]. The cell walls of the transfer cells (modified aleurone cells in the crease region) contain less FA, and PCA is present in negligeable amounts. Moreover, their AXs are more substituted (A:X = 0.6), and they contain higher amounts of BGs once they mature [31]. Thus, the origin of the aleurone cell wall is related to its function and impacts on the layer’s technological and health properties.

However, some minor components, such as proteins, can be found in the cell walls of the aleurone layer, cross-linked to AXs, BGs or sometimes to hydroxycinnamic acids [9,52]. They amount to about 1% of the total aleurone cell walls and show a similar amino acid composition as the proteins found in the endosperm (Table 3). These proteins can be classified as either glycine (37–86%), proline (11–39%), or serine rich (up to 23%) [52].

Compared with other bran layers, the aleurone cell-wall comprises low, if any, amounts of cellulose, glucomannan and lignin (Table 1) [10,25,44,53].

**Table 3 foods-11-03552-t003:** Proteins and amino acids repartition in the wheat aleurone layer, bran and whole grain.

	Aleurone Layer *	Bran *	Whole Grain
Protein (%, *w*/*w*) Part in tissue Part in cell-wall Total in grain	≈30.0 [32,33,54,55,56] 1.0 [36,37] ≈15.0 [32,33,54,55,56]	15.2–16.9 [3] 9.2 [36] 14.0 [3]	10.0–15.0 [32,33,54] - - -
	AMINO ACIDS		
	(in %) [54,56]	(in g/16 g N) [54]	
Alanine	5.9	4.9	3.5
Arginine	11.1	7.0	4.6
Aspartic acid	7.9	7.2	5.0
Cysteine	-	2.0	2.2
Glutamic acid	20.9	18.6	30.6
Glycine	5.8	7.1	3.9
Histidine	3.4	2.6	2.2
Isoleucine	3.6	3.5	3.8
Leucine	6.5	6.0	6.7
Lysine	4.8	4.0	2.7
Methionine	1.6	1.6	1.7
Phenylalanine	3.8	3.9	4.6
Proline	6.3	5.9	9.8
Serine	2.9	4.5	4.8
Threonine	2.9	3.3	2.9
Tryptophan	4.0	1.6	1.2
Tyrosine	3.3	2.8	3.1
Valine	5.3	5.0	4.7

* The aleurone layer and the bran amount respectively to 5–8% (*w*/*w*) and 5.5–9.5% (*w*/*w*) of the wheat grain [21]. All data were placed in the same unit to facilitate comparison.

#### 2.2.2. Intracellular Medium

The intracellular medium represents 70% of aleurone cell dry mass [19]. It is composed of a large nucleus and aleurone grains (or granules). The latter are inclusion bodies, or vacuolar units of two types with a spherical structure (2–4 µm in diameter), surrounded by non-polar lipid droplets. Type I inclusions comprise phytic acid crystals. In those, dihydro-phosphate traps minerals, including calcium, magnesium and zinc, by chelation, forming phytate complexes. This chelation of minerals results in a decrease in their bioaccessibility, which subsequently reduces the nutritional value of the aleurone intracellular medium unless they are released, for instance by the actions of endogenous enzymes (phytase) [9,57].

The other type of aleurone grain (type II) contains niacin (B3 vitamins) and proteins [9,10,58,59,60]. The proteins contained in type II inclusions represent 15 to 20% of total wheat grain proteins (dry mass) [5,61]. They are mostly storage proteins (mainly 7S globulins), despite the presence of numerous metabolic and defence enzymes [62]. The presence of lysine in these proteins is beneficial as it is a limiting amino acid in cereal products [9].

Overall, the aleurone grains provide numerous minerals accounting for 40% of total grain minerals, such as phosphate (80% in phytate form), magnesium, manganese, iron, potassium and sodium [10,11], as presented in Table 4. They are also a source of vitamins (Table 5). Indeed, thiamin and riboflavin are mostly present in this bran layer. Moreover, it is a source of lignans, especially syringaresinol, which are renowned for their antioxidant properties [10]. Significant levels of carotenoids (lutein and zeaxanthin) can also be found [63], as well as betaine and choline, which are twice as concentrated as they are in bran [64], as presented in Table 6. These bioactive components contribute to the nutritional value of the aleurone layer, and demonstrate the potential benefits of its addition to cereal products for the consumer.

In addition, the aleurone intracellular medium is rich in endoplasmic reticulum, mitochondria, membrane-bound vesicles and lytic vacuoles [26]. No starch granules are present in the aleurone cells despite their proximity to the starchy endosperm [51].

In both cell walls and intracellular medium, most compounds are susceptible to great variations in concentration. Indeed, the effects of genetics, cultivar, and culture and storage conditions must be considered, as they can influence the amount of aleurone’s compounds, thus having repercussions on its technological, health and nutritional properties [11,69]. For instance, coloured wheat varieties will differ greatly in their phytochemical composition [70,71,72].

## 3. Health and Nutritional Benefits

As presented, the aleurone layer is a source of many bioactive compounds that can potentially exhibit nutritional and/or health benefits for consumers. The benefits related to the layer’s composition are impacted by the natural variability of the components. Moreover, as the following studies have been performed on materials of different purity and in various forms due to non-identical transformations and processes of obtention, the concluding remarks must then be approached as potential beneficial effects, as they are related to the specific conditions of materials and processes.

### 3.1. Digestibility and Colonic Fermentation

When ingested, the aleurone seems to have better digestibility than the other bran layers, with an overall digestibility value of 30%. This might be related to the absence of lignin in the aleurone, which is known to impair digestion [10,20]. However, this layer shows very poor digestibility in the upper digestive tract, especially when it comes to mineral absorption [57]. Furthermore, the release of minerals from the food matrix may be hindered by the presence of DFs, which cannot be digested in this part of the human gastrointestinal system [9]. Their bioaccessibility is even more decreased due to the presence of phytates, by which they are complexed [9].

In contrast, the in vitro colonic fermentation potential of aleurone seems to be better than that of wheat bran [73]. In multiple studies, the consumption of aleurone led to the stimulation of microbial activity (*Bifidobacteria dorea* and butyrate-producing *Roseburia* spp.) in the caecum and colon, thus leading to a higher yield of propionate and butyrate. These important short-chain fatty acids (SCFA) are renowned for their health benefits, namely in cancer prevention [57,74,75]. A decrease in health-detrimental bacteria, such as *Bilophila*, *Escherichia* and *Parabacteroides*, was also reported [75].

These observed effects probably arise from the presence of DFs in the aleurone layer. For instance, AXs in the layer’s cell walls can exert beneficial health effects by inducing the proliferation of healthy gut microbiota and providing a substrate for increased SCFA production. Additional effects could include increased faecal bulk and viscosity, accelerated transit time and potential binding to cancer-inducing molecules [34,35]. These prebiotic effects could be accompanied by the cholesterol-lowering activity of AXs [76]. Slightly branched AXs are more easily degraded than AXs with a higher substitution degree, as they ferment more slowly or remain unfermented [57]. As the aleurone layer’s AXs are poorly cross-linked, this might be beneficial when compared with other bran layers.

Nevertheless, the presence of DFs can also hinder the bioavailability of other bioactive compounds present in the aleurone layer by decreasing their bioaccessibility. Since most fibres in this layer are either only partially or not digested by the intestinal flora, most of these important compounds cannot be released for absorption [9]. This includes the absorption of minerals complexed with phytate. However, DF fermentation in the colon can increase minerals’ colonic absorption by increasing the production of SCFA, in turn reducing the pH in the intestine, thus solubilising phytic complexes and releasing minerals [77].

### 3.2. Health Benefits of Aleurone Consumption

In various clinical studies, a diet rich in wheat aleurone has been shown to reduce the risk of cancer, cardiovascular disease, obesity and diabetes [12,78]. Symptoms such as hypertension and hyperglycaemia reduction have been reported after consumption of aleurone in the long term, which contributes to the prevention of obesity and hyperlipidaemia [79]. This can also be seen with a decreased low density lipoprotein (LDL) cholesterol level post-ingestion [80]. Moreover, aleurone ingestion seems to reduce the risk of colon cancer after fermentation in the colon by decreasing secondary bile acid production, inducing apoptosis and cell differentiation, as well as detoxification [81,82,83,84].

In human studies, a moderate intake of aleurone raised red blood cell folate levels, decreasing plasma homocysteine levels. This effect is beneficial since high plasma homocysteine is a risk factor for both cardiovascular disease and cancer via DNA damage [85,86,87]. Moreover, the authors concluded that the folate contained in the aleurone was then highly bioavailable and bioefficient [86].

However, Keaveney et al. [88] reported contrasting results, as the consumption of 50 g of a wheat aleurone fraction from Bühler A.G. increased the betaine plasma concentration by 2.5 times, but neither plasma tocopherols, folate nor choline levels were increased. Another study confirmed this effect of aleurone on metabolic risk factors; higher plasma betaine and related lower plasma homocysteine were observed [80]. Nonetheless, the most recent study reported that upon consumption of 27 g of aleurone daily by overweight and obese subjects, no significant changes were observed in health-related biomarkers, including plasma homocysteine and SCFA levels [89].

Studies on the effect of wheat aleurone consumption on inflammation markers have reported a decrease in pro-inflammatory tumour necrosis factor (TNF)-α (one of the most important cytokines in the immune system [90]) in LPS-stimulated U937 macrophages [91]. Moreover, a decrease in C-reactive protein (CRP) was also noticed (a biomarker of inflammation used in the prediction of coronary heart disease [92]), probably via a reduction in LDL-cholesterol levels. This could imply that aleurone consumption may change hepatic metabolism either by the action of its independent bioactive components or by their interaction [93].

Although the health benefits observed upon aleurone consumption in clinical studies are numerous, they are still hypothetical. As the studies did not involve the same test conditions and used different materials, they are thus not comparable. Moreover, it does not consider the natural variability in the composition of the aleurone layer, nor does it wholly consider the matrix effects of its consumption in a normal diet with other food products. In conclusion, the observed health benefits may arise upon aleurone consumption, but they should be referred to as potential effects.

### 3.3. Antioxidant Capacity

In addition to the aforementioned beneficial health effects, aleurone is also a source of many bioactive compounds with antioxidant activity. It is the bran layer with the highest total antioxidant capacity and can provoke a prolonged anti-inflammatory effect after consumption [91]. This is partly due to its high phenolic acid content and mainly allotted to FA, which accounts for 60% of the aleurone antioxidant capacity [45].

The consumption of 50 g of this milling fraction from Bühler A.G. has also been reported to increase the FA plasma concentration [94]. Furthermore, the latter phenomenon seems to be related to enhanced FA bioavailability [95], and more generally an increased phenolic acid bioavailability, which is apparently conserved even when the aleurone fractions are incorporated into bread [96].

Due to its resonance-stabilised phenoxy radical structure, FA can inhibit lipid peroxidation via superoxide scavenging [97,98]. Moreover, the presence of a methyl group in C3 enhances resonance stabilisation, thus making it a very stable antioxidant that does not initiate an oxidative chain reaction [99,100]. Although this activity is increased when FA is in its dimerised form (whereas it is mostly in monomer form in the aleurone layer, Table 2), it still exhibits better LDL inhibition than ascorbic acid, a powerful antioxidant [9,101].

However, the bioavailability of FA is decreased when in this dimerised form or if ester-linked to AX, the latter amounting to the majority of FA in the aleurone layer [45,73,102]. The low degradation of DFs by the microbiota could present better health effects than a high peak after ingestion since phenolic compounds are continuously released in the plasma [103]. Moreover, the bioavailability of such compounds could be improved by the addition of cell wall degrading enzymes that release them [104].

In addition to FA, other phenolic acids (mainly PCA and sinapic acid) also contribute to the total antioxidant capacity of the aleurone layer [12]. Bioactive components, such as phytate, phytoestrogens (including lignans), and anthocyanins, are also involved [77,105,106,107]. Vitamin E participates equally by quenching singlet oxygen and nitrogen-oxide radicals. More precisely, α-tocopherol works as part of an antioxidant network in breaking lipid-soluble chains [108]. Vitamin B6 also possesses antioxidant activity and thus participates in protection from oxidative stress [109]. The combined effects of these compounds present in the aleurone layers make them a valuable source of antioxidants, provided they retain this property after transformation (i.e., breadmaking).

### 3.4. Bioactive Components of Aleurone and Related Potential Nutritional and Health Benefits

The aforementioned effects have been reported for the consumption of aleurone as a whole, but the health benefits of the bioactive components it contains have also been investigated separately.

The presence of vitamins in the aleurone layer (Table 5) can be beneficial from both nutritional and health points of view. For instance, tocopherols are involved in the regulation of cell signalling and gene expression and are also known to delay the progress of degenerative diseases. The other type of E vitamins, tocotrienols, are involved in the prevention of neurodegeneration. They can also induce immune responses as well as lower cholesterol and prevent cancer [108]. Unlike E vitamins, folate (vitamin B9) is renowned for its biological activity in normal foetus neural tube development [11]. In addition, niacin is involved in carbohydrates and fats metabolism [110].

However, anthocyanins may exhibit anticancer properties [106,107], and lignans may lower cholesterol and show potential estrogenic activity [12,105]. Moreover, the benefits of betaine include its role as an osmolyte, remethylating total homocysteine, therapeutic agent for non-alcoholic fatty liver disease, and lipotrop [80].

Lastly, the aleurone layer contains an essential amino acid: lysine (Table 3). Its presence is beneficial for the end-product nutritional value since lysine is a limiting amino acid in cereal grains [9]. In addition, the presence of arginine in aleurone proteins can also be valorised due to its role in vascular dynamics and endothelial function, which can improve [12].

As these compounds are present in the aleurone layer, this adds to its value, provided that they remain present in the end-product and conserve their potential health-benefit properties through the process.

## 4. Potential of the Aleurone Layer as an Ingredient in Bread- and Cereal-Based Products

### 4.1. Extraction of the Aleurone Layer and Its Challenges

The aleurone layer’s potential can only be revealed if it is first extracted. However, no universal process yet exists. In addition, as the grain’s milling properties (friability) are dependent on its constituents, themselves related to the culture conditions and genetic background, the process must constantly adapt to the raw material. Therefore, this represents a major hurdle in the utilisation of the aleurone layer and in the exploration of its properties.

Another challenge arises from its composition. The aleurone layer, although botanically part of the endosperm, is considered by millers as belonging to the bran fraction. Since it is tightly adhered to the pericarp, it is usually removed from the endosperm during conventional milling. This tight adherence to seed coats also makes it difficult to separate the aleurone from the rest of the bran [10,12]. Consequently, most existing processes for the retrieval of the aleurone layer start with bran material. Multiple procedures have been patented [14,111,112,113,114] and two companies have mainly been known to produce and commercialise aleurone-rich flour: Bühler A.G. and Cargill Limited (through Horizon Milling with the GrainWise brand). However, it seems that most existing processes result in the obtention of aleurone-rich flour that is either not highly concentrated [13] or low yielding [14]. A summary of the composition of aleurone-rich fractions issued from the existing processes mentioned below, available in the literature, is shown in Table 7. As shown in the table, the purity of the obtained aleurone-enriched fraction depends greatly on the extraction process performed, which explains the discrepancies observed in their composition when compared to that of the pure hand-isolated aleurone layer. These differences in composition may also arise from the use of different analytical methods among publications.

Most of these processes extract the aleurone layer from bran components by dry-fractionation, a succession of mechanical or physical unitary steps [10,115,116]. Many researchers first aim to dissociate the different bran tissues, which can be conducted by grinding. They include a separation step that then enables sorting out the particles according to their size, mass, density or electrostatic properties [116]. The obtained milling fraction can thus be added to basic wheat flour to enrich it with the aleurone layer [115].

The aleurone layer is extensible, similar to the intermediate strips of wheat bran, and it has an elastoplastic rheological behaviour. Its mechanical characteristics are impacted by the degree of feruloylation of its AX, particularly by the presence of FA dehydrodimers [44]. Hence, the mechanical stress generated by dry fractionation processes first affects the aleurone cell walls, which crack, allowing the cell contents to be released [117]. According to Rosa et al. [118], the velocity of phytic acid release could thus be used as a marker to estimate aleurone cell opening.

Electrostatic separation of aleurone from other bran tissues is an interesting process since the aleurone layer presents unique electrostatic properties compared to the other strips. However, this process can be influenced by multiple parameters, such as particle size, composition, microstructure and moisture content [68].

The main advantage of physical or mechanical extraction methods is that they do not require the use of chemical products that can interact with the matrix and decrease the product’s purity and phytochemicals’ functionality [116]. Compared with wet processes (chemical and enzymatic treatments), they also enable higher energy efficiency [10,116]. However, the succession of unit operations may impact the antioxidant and secondary metabolites of the aleurone layer [3]. Moreover, grinding generates various particles from bran tissues of different sizes and densities, which are hard to differentiate, hence the reported end-product’s low purity [116].

This type of process was used in the patented method by Stone and Minifie [14], who first used hammer-milling in wheat bran containing 34% of aleurone cells, followed by sieving, electrostatic fractionation, and a final separation through an electric field. A 95% purity of aleurone cells was obtained with a 10% yield [14,68]. Nonetheless, alternative methods exist: humidification then micro-grinding of wheat bran with a friction roller mill [114]; sequential pearling cycles in a vertical abrasive polishing machine [74]; centrifugal impact milling [68]; ultrafine grinding and electrostatic separation [119]. Different outcomes have been reported with these processes, with varying aleurone purity and yield.

However, there are limited studies on the extraction of the aleurone layer by wet processes (chemical and enzymatic treatments). For instance, the maceration of wheat bran in chemical reagents, such as organic solvents, has been tested but not in an industrial scale [10].

Nonetheless, dry and wet processes can be coupled. A patented method isolated aleurone by successive steps of cleaning, steaming, stabilising, roller-milling, sieving, fine grinding and air-classifying. However, the end-product still contained 36.5% starch, demonstrating low purity [87,120]. In addition, the patent deposited by Kvist et al. [121] subjected wheat bran to several enzymatic treatments, wet milling steps, sequential centrifugation, and ultrafiltration. Other researchers coupled successive steps of milling, sieving, air classification and centrifugation with benzene-carbon tetrachloride mixtures at laboratory scale [10].

Although many experiments have been conducted and sometimes patented to extract and isolate the aleurone layer, the challenge of measuring end-product purity has arisen. Thus, researchers have defined biochemical markers to differentiate grain parts. These biochemical markers can be used to determine the extent to which the aleurone layer is extracted from other grain components. Starch, phytate, p-coumaric acid, alkylresorcinols and FA trimer are used to estimate the proportion of the endosperm, aleurone cell content and cell-walls, intermediate layer and outer pericarp, respectively [46,65,122]. However, the relative amount of grain tissue can only be calculated when compared to the reference values. The latter were values of the same parameters from pure isolated tissues of identical wheat cultivars. Thus, it limits their utilisation for characterisation since pure fractions are obtained from hand-isolated tissues, a long and labourous process. Moreover, these markers are susceptible to natural variability among wheat cultivars due to the culture conditions and genetic background [50,123]. As an alternative to these biochemical markers, microscopy analyses can be performed to estimate the purity of the extracted fractions [115].

**Table 7 foods-11-03552-t007:** Composition of aleurone-enriched fractions issued from different extracting processes in literature.

Origin of Products *	1a	1b	2	3	4
ASH (g/100 g)	9.3–13.3 [12,124,125]	≈ 10.0–13.3 [7,12,126,127]	4.1 [85]	7.2–7.34 [128,129]	3.9–5.1 [130,131]
MOISTURE (g/100 g)	-	8.0 [7]	5.4 [85,87]	8.14 [128]	6.7–8.98 [130,131,132]
CARBOHYDRATES					
Arabinoxylans (g/100g)	-	24.3 [127]	-	-	14.39 [131]
A:X ratio	0.62 [133]	0.35–0.46 [7,127,133]	-	-	-
β-glucans (g/100 g)	3.4 [133]	3.91–4.5 [127,133]	-	-	1.7 [132]
Cellulose (g/100 g)	10.6 [133]	6.0 [133]	-	-	-
Pentosans (g/100 g)	-	-	-	-	20.2–21.6 [134]
Starch (g/100 g)	1.9–5.8 [124,133]	2.2–5.8 [126,127,133]	36.5 [85,87]	2.5–12.75 [128,129]	33.46 [130]
Total Dietary Fibers (g/100 g)	39.7–60.0 [12,124,125,133]	39.7–49.2 [7,12,126,127,133]	15.4 [85,87]	43.36 [128]	27.90–44.3 [130,131,132]
Soluble Dietary Fibers (g/100 g)	4.1 [125]	-	-	3.07 [128]	-
Insoluble Dietary Fibers (g/100 g)	50.0 [125]	-	-	40.15 [128]	-
MINERALS AND TRACE ELEMENTS					
Total (g/100 g)	5.8–9.8 [124,125]	7.0–9.8 [7,126,127]	6.5 [85,87]	4.5–6.33 [128,129]	4.1–4.7 [134]
Calcium (Ca) (mg/100 g)	76.2 [125]	93 [10]	-	-	-
Copper (Cu) (mg/100 g)	-	-	-	1.35 [128]	-
Iron (Fe) (mg/100 g)	21.3 [125]	26 [10]	-	13.93 [128]	-
Magnesium (Mg) (mg/100 g)	690–800 [12,125]	850–1030 [10,12]	-	770 [128]	-
Manganese (Mn) (mg/100 g)	-	-	-	12.7 [128]	-
Sodium (Na) (mg/100 g)	1.7 [125]	-	-	-	-
Phosphorus (P) (mg/100 g)	1900 [125]	2540 [10]	-	1857 [128]	-
Potassium (K) (mg/100 g)	1900 [125]	2250 [10]	-	1780 [128]	-
Zinc (Zn) (mg/100 g)	11.4 [125]	14.0 [10]	-	12.05 [128]	-
PHENOLIC COMPOUNDS					
Total phenolic acids (mg/100 g)	-	-	-	-	457 [132]
Total hydroxycinnamic acids *Free* (mg/100 g) *Bound* (mg/100 g)	1.28 [135] 47.58 [135]	1.22 [135] 60.65 [135]	-	-	-
*p*-Coumaric acid *Total* (mg/100 g) *Free* (mg/100 g) *Bound* (mg/100 g) *Conjugated* (mg/100 g)	- - 0.60 [135] -	16.0 [126] 1.0–1.5 [126] 0.99–1.0 [126,135] 0 [126]	-	-	-
Sinapic acid—bound form (mg/100 g)	0.46 [135]	0.53 [135]	-	-	-
Alkylresorcinols (mg/100 g)	1107 [135]	993.24 [135]	-	-	138 [132]
Flavonoids (mg/100 g)	9.65 [135]	8.95 [135]	-	-	-
Lignans (mg/100 g)	6300 [133]	4700 [133]	-	-	-
Phytic acid (mg/100 g)	6900 [125]	-	2360 [85,87]	-	-
PROTEINS					
Total (g/100 g)	13.3–18.0 [12,124,125]	21.0–22.2 [7,12,126,127]	23.6 [85,87]	15.2–17.46 [128,129]	16.5–21 [130,131,132]
VITAMINS					
Total (mg/100 g)	>29.0 [125]	30.0 [126]	-	-	-
Thiamin (B1) (mg/100 g)	0.87–1.6 [12,125]	1.1–1.4 [10,12,126]	-	1.26 [128]	-
Riboflavin (B2) (mg/100 g)	0.3 [125]	0.2–0.3 [10,126]	-	0.32 [128]	-
Niacin (B3) (mg/100 g)	24.0 [125]	21.0–32.9 [10,126]	-	22.87 [128]	-
Pantothenic acid (B5) (mg/100 g)	-	5.0 [126]	-	1.87 [128]	-
Pyridoxin (B6) (mg/100 g)	0.3 [125]	1.3–1.4 [10,126]	-	2.52 [128]	-
Folate (B9) (mg/100 g)	0.8 [125]	0.1–0.2 [10,126]	-	0.2 [128]	-
Tocopherols and tocotrienols (E) (mg/100 g)	2.0 [125]	0.8–1.2 [10,126]	-	-	-

* Origin of products, as listed below. **1a**: Bühler A.G. (55–70% aleurone purity); **1b**: Bühler A.G. (75–90% aleurone purity)—patented method [111]; **2**: Goodman Fielder Milling and Baking Pty. Ltd. (90% aleurone-rich flour with 10% of waxy maize starch); **3**: Cargill Limited and Horizon Milling (Grainwise); **4**: Jiaxing Zhishifang Food Science and Technology Co. (Shandong, China), 14% wb. All data were placed in the same unit to facilitate comparison.

### 4.2. Application to Breadmaking

#### 4.2.1. Aleurone Bread Nutritional Profile

According to past reviews and experiments, there are many benefits to incorporating an aleurone-rich flour into bread and bakery products, starting with an improved nutritional profile of the end-products. This amelioration is related to increased DF and protein (mainly albumin and globulin) content at the expense of readily digestible carbohydrates [13,15,119,131,136]. The enhancement of minerals, including phosphate, magnesium, manganese and iron, and bioactive compounds such as phenolic acids, antioxidants, phytoestrogens and sterols, also increase the value of the obtained end-products [16,136]. This improved composition confers the end product a nutritional profile similar to that of whole wheat products [128], while equally making it a good source of fibre [12]. However, the nutritional benefits of aleurone-rich products are accompanied by increased phytate content, which is known for its antinutritional effect [57].

#### 4.2.2. Aleurone Bread Dough Characteristics

Despite these beneficial nutritional properties, the incorporation of the aleurone layer for breadmaking leads to changes during dough formation, which affects the sensory attributes of the end-product. With its high DF content (Table 1 and Table 7), the aleurone layer impairs dough hydration properties. The AXs and BGs contained by the aleurone layer compete for water with the proteins forming the gluten network, thus increasing the water absorption of the dough and retarding the dough development time [136,137]. The water retention capacity is also affected by fibres that take up a large amount of water (3.5 to 6.3 times their weight for WEAX and 6.7 to 9.9 for WUAX) by binding through hydroxyl groups, resulting in a longer mixing stability due to the alteration of the gluten structure [6,131,136].

In addition, the presence of these fibres has a diluting effect on the starch granules. Damaged starch content is then decreased as well as the falling number. The latter effect is further reduced by the increase in α-amylase activity in the presence of calcium. Indeed, this metalloenzyme requires calcium for its performance, which is provided by the aleurone layer (Table 4 and Table 7). In addition, these observed properties seem to increase in relation to the aleurone-rich flour dosage [136].

The effect of aleurone incorporation on starch also influences the pasting properties of dough. Multiple studies have shown a decrease in peak viscosity, as well as in the retrogradation of dough [131,132]. This might not only stem from the presence of fibres that interfere with starch granule swelling and increased amylase activity but also from the combined effect of other aleurone constituents. For instance, the presence of fat and FA can also impact pasting properties, in addition to an already low starch content [132,138].

Nevertheless, the aleurone dough exhibits higher Rapid Visco Analyzer (RVA) parameters, revealing a strong gel ability greater than that of whole wheat flour [131]. According to Bucsella et al. [136], this could be due to the swelling of fibres, which form a strong gel despite the lower starch content. This gel is described as being more resistant to heat and mechanical stress.

The gluten network can also be strengthened following the addition of aleurone-rich flour to bread dough (up to 40%). However, according to Mixolab (Chopin) measurements, the dough development time is increased due to the presence of fibres that compete for water and hinder gluten network formation by intercalating between the proteins, resulting in a more heat-stable and stress-resistant dough [131].

This increase in dough stability is also depicted by firm elastic-like behaviour due to the stronger gluten complex [131,136]. The increased protein content (albumin) and the strengthening effect of AX binding to gluten via the oxidative dimerization of FA also contribute to these observed effects [139]. Instrumentally, this translates into an increase in dough stability and break time, as well as delayed weakening [136].

Despite the aforementioned beneficial traits observed due to the aleurone components, most of them are dose dependent. Excessive addition of aleurone-rich flour to the dough can lead to deleterious effects on dough rheology.

#### 4.2.3. End-Product: Aleurone Bread Characteristics

The addition of aleurone-rich flour to bread-making has an impact on end-product quality, although the results of the researcher’s findings are contradictory. This may be related to the aleurone-enrichment level, the purity of this material as well as the bread formulation process, compiled in Table 8.

Some report a decrease in loaf volume, accompanied by reduced height and increased weight [13,15,16,17]. For instance, Bagdi et al. [15] observed a diminution of 27% of the specific volume for a bread prepared with 100% of aleurone-rich flour compared to a control white bread, as well as a reduction in loaf height of 13%. Using the same breadmaking process (ICC Standard Method 131), Bartalné-Berceli et al. [16] obtained a height decrease of 15 and a 7.2% weight increase with a bread containing 25% of aleurone-rich flour compared with a control white bread. These tendencies are further incremented with a higher aleurone flour input (50%), where almost half of the height was decreased and 3.6% of the weight was increased compared with the control.

Other studies have observed a higher loaf volume than white bread upon aleurone incorporation or have found insignificant changes. The texture in these experiments was also reported to be softer than white bread, which means that the crumb was less dense [12,136,140]. Indeed, Tian et al. [140] described an increase of 40.91% in bread specific volume using aleurone-rich flour (modified GB/T 35869-2018 procedure with 54.11% of aleurone layer content). However, this beneficial effect could be attributed to the presence of hemicellulases (at 40 mg/kg) that enable the formation of WEAX from WUAX. Similar results were obtained from breads made with a sourdough preparation (MSZ-6369-8:1988) incorporating an aleurone fraction, even though the observed volume increase was not significantly different from that of the control bread [136]. Breads made from straight dough and sponge dough processes with additives containing aleurone-rich flour (20%) also show this beneficial trait [12].

Overall, it seems that these beneficial effects could be related to the presence of hemicellulose-degrading enzymes—either endogenous (sourdough) or exogenous (as an additive)—or additives or a special breadmaking process, each enabling the revelation of the aleurone layer’s full technological potential.

Unlike for the volume and texture of the breads, the appearance of the end-product is equivocal: the crumb colour (measured by colorimetry) is darker than white bread, even brownish, which can be a limiting factor for some consumers. However, it is still lighter than whole wheat products [13,15,57].

As for the taste of the bread, diverging results also occur. Whereas some report a flavour similar to that of white bread, especially when a long fermentation process takes place [12,128]; others describe a bread that is more bitter and sour, even rye-like [13,15]. In addition to the last finding, Amrein et al. [57] outlined a gritty mouth-feel, which is a limiting factor for the consumers of the study. However, the smell of the products is reported to be more intense and sour [13,15].

Overall, bread made of aleurone-rich flour in different proportions showed similar properties to that of white breads but with the nutritional profile of whole wheat breads. More thorough experiments on the relevance of aleurone addition in cereal products compared to other wheat kernel layers should be conducted, as the diversity of breadmaking methods and starting materials is great in the existing studies. Nevertheless, the results are still contradictory and lead to either a decrease or increase in end-product consumer acceptability. The use of special breadmaking technologies or additives, such as cell wall degrading enzymes, could thus reveal the aleurone layer’s full technological potential.

#### 4.2.4. Underlying Mechanisms

Many of the adverse or positive technological effects due to the addition of aleurone-rich flour to bakery products stem from its unique composition and, more specifically, its high protein and DF content. Indeed, studies investigating the effect of DF, AX and bran incorporation into bakery products showed similar properties to those described in aleurone-enriched products.

Most experiments on this subject describe that the addition of fibres to bread dough increases dough development time, water absorption and strength. However, it also seems to weaken the dough’s tolerance to mixing and fermentation [141]. This results for most studies in a reduction in loaf volume [5,141,142,143,144], an increase in crumb firmness [142,143,144], and a darkened crumb appearance [141,142,143,144].

Hypotheses exist to explain the mechanisms underlying these results, which corroborate those of aleurone-rich products. First, fibres with their high water binding capacity might compete for water with starch and gluten, thus keeping wheat proteins from sufficient hydration for the formation of the gluten network [5,141,142,145,146,147,148,149]. Another explanation is that fibres dilute gluten, thus affecting its gas-holding capacity [5,141,142,145,146,147,148,149]. Nonetheless, this impairment in gas retention that causes a loss of loaf volume could also be due to the shortened and lowered resistance to dough extension upon DF addition, which increases the concentration of soluble cell wall materials and disrupts the gluten network [142,150].

In addition to the previous mechanisms of action that hint at a physical mode of action, a chemical hypothesis also exists that states that FA linked to DFs could mediate AX–AX and AX–protein cross-linking (through FA–tyrosine linkages), thus impacting gluten properties [146,147,148,149]. This would be possible in the presence of oxidants or enzymes (such as laccase and peroxidases) that provoke the dimerization of FA, thus creating covalent linkages between AX chains. Moreover, this dimerization increases the water retention capacity of AXs, which directly affects the gluten network [6].

More specifically, studies conducted on the addition of AXs to bakery products could be helpful in understanding the mechanisms underlying the aleurone-enriched bread properties, since they represent a large part of this layer. Similar to the addition of general DFs, an increase in the water absorption of the dough is reported due to the high water retention capacity of AXs, which increases dough consistency [151]. According to Berger and Ducroo [6], to reach the same dough consistency as the control dough on the Brabender farinograph, 0.5 to 2% of additional water should be incremented per percent of AX supplemented.

As for the negative effects on gluten network formation due to AX addition, they could stem from the steric hindrance of the increased batter viscosity that limits components mobility, thus decreasing the formation of gluten aggregates and starch entrapment in its matrix [151]. Nonetheless, the observed effects are not as important as the extent of their addition and their molecular size, but most importantly, depend on the breadmaking quality of the initial flour used for the experiments [152].

Furthermore, the water extractability of AXs is also a determining factor in the adverse effects it causes on bread and bakery products. For instance, water-unextractable AXs (WUAX) seem to generate more deleterious side effects upon their addition than water-extractable AXs (WEAX). This might explain the contradictory results with aleurone applications since it mainly contains WUAXs, which can be transformed into WEAXs during breadmaking.

The use of WUAXs in bakery products is often reported with breads of lower volume, coarser crumb and higher firmness [39]. To explain this phenomenon, there are three hypotheses: (i) WUAXs form physical barriers for wheat proteins during dough development [39]; (ii) these AXs form intrusions in the gas cells during fermentation [39,153]; (iii) the WUAXs compete for water with the gluten network, thus impairing its formation and leading to a fracture effect that increases dough resistance to extension [6,7,39]. The latter hypothesis is believed to be more accurate since a correction of dough hydration (2% per percentage of AXs added) improves its extensibility [6].

In contrast, the use of WEAXs in bread dough yields contradictory results, even though they are mostly beneficial. Globally, a finer and more homogenous breadcrumb is depicted, with a bread that is softer [39]. The loaf volume is also impacted, but contradictory results are obtained. The observed higher volume is usually obtained with the use of high molecular weight WEAXs [39,153,154]. The underlying mechanisms explaining these effects include an increase in liquid film stability and thus in dough aqueous phase viscosity [7,39]. Moreover, WEAXs of higher molecular weight can form a secondary network, weaker than gluten, which enforces the latter and stabilises it through the dimerization of FA and by physical entanglement, either of gluten or between WEAXs [39]. By increasing the dough’s gas retention capacity, the resulting breads become higher [6]. According to [6], the higher the WEAXs molecular weight, the highest beneficial effect is observed.

### 4.3. Application to Other Food Products

Although the incorporation of aleurone-rich flour into bakery products have mostly been studied, its application to other product categories exists. For instance, Cargill Limited [128] developed cereal flakes and extruded snacks with 35% of aleurone-rich flour, as well as high-protein bars containing 20%. Ready-to-eat cereals enriched in aleurone were also studied by Byrne [155].

Other applications entail aleurone-enriched pasta and noodles. Whereas the former was described as healthier than wholemeal spaghetti due to higher protein, fat, and DF content, its consumer acceptance was decreased. Although it showed improved quality characteristics (lower water uptake, higher cooked pasta firmness, higher tensile strength and lower stickiness), the darker, more intense, bitter and sour taste of the pasta influenced consumer appreciation of the product [138]. The second used a combination of aleurone-rich flour and transglutaminase, which resulted in noodles with less cooking loss and the best sensory evaluation when compared to traditional noodles [130].

Finally, Yang et al. [134] used aleurone-enriched flour and cell wall degrading enzymes for the production of Chinese buns. They found that the action of enzymes promoted the WEAX content while also increasing the water availability to the gluten-forming proteins, resulting in softer dough, especially when enzyme activities were combined.

## 5. Optimization of the Aleurone Layer’s Potential

Despite the aleurone layer’s nutritional and potential health-benefit properties, its use on the market is scarce. This might be related to the hurdles in its extraction, as well as the rheological issues it has with the end-product. To overcome the latter and obtain a product with improved nutritional traits without alteration of its technological properties, aleurone-rich flour can be subjected to different processes, whether physical, chemical or biological. By modifying the aleurone constituents and, most specifically, its DFs, which are mainly responsible for the observed adverse effects, the technological potential of the aleurone layer could thus be optimised. Moreover, these processes could also enhance the health benefits associated with the intake of this bioactive milling fraction.

Many physical treatments, whether thermal, non-thermal, dry or wet, focus on the particle size reduction in the material, as it can lead to many beneficial effects. It includes an improved antioxidant and bile acid-binding capacity, a greater bioavailability of phenolic compounds and vitamin E, a higher production of colonic SCFA, and a faster digestion rate, which can increase transit time and decrease faecal bulking [13,19]. For this purpose, different techniques can be performed. Whereas milling refers to the process of separating the endosperm (known as white flour) from the bran (outer layers and germ) [156], grinding uses shear stress and compression to reduce the particle size [157]. Both can be used to obtain a particle size below that of aleurone cells (around 50 µm) to release their content [117,158].

The results of those studies concluded that as the particle size of the material decreased, phytate extractability was enhanced, and phenolic acids were released [117], thus improving the mineral bioaccessibility of aleurone [158]. The hydration properties (namely water holding and binding capacity) of the milling fraction are also reduced, which subsequently negatively affects its fermentability [127]. In addition, an increase in conjugated and free FA post-grinding has been observed, as well as the release of aleurone intracellular compounds (soluble proteins, vitamin E and phytic acid). The combined effects of improved bioaccessibility of antioxidant compounds, as well as the greater exposure of phenolic moieties, result in an enhanced antioxidant capacity of the modified aleurone ingredient [9,118,126,127,159].

In addition to physical treatments, the use of chemical processes to modify, inter alia, the solubility of fibres is of equal interest for the improvement of the aleurone layer’s technological potential. Their benefits lie in the fact that they are the only ones that can provoke a polymerisation of the materials. However, the final product can be of low purity, with a high degree of hydrolysis and modified functional groups [160,161].

Experiments conducted by Bagdi et al. [162] revealed that hydroxyl radical treatment, •OH oxidation and cross-linking of AXs extracted from aleurone-rich flour modified its bile acid-binding capacity and could even enhance the cholesterol-lowering effect of AXs. Besides these improved health effects, the use of an alkaline treatment on the aleurone’s WUAXs could be beneficial, as it was reported to maintain the fibre’s molecular weight while increasing its water solubility [154]. Moreover, as carboxymethylation of wheat bran enhanced its health properties (increased total antioxidant capacity, total reducing power, Fe^2+^ chelating capacity and DPPH radical scavenging capacity), the use of this chemical mean of treatment on the aleurone layer could equally raise its value [163].

Nonetheless, most of the treatments performed on aleurone-rich flour in the literature have been conducted by biochemical means. Rhodes and Stone [124] studied the effect of combined methods, namely ultra-fine grinding and enzymatic treatment with xylanase and feruloyl esterase. Upon those treatments, changes in the aleurone layer structure were observed, as well as an increase in free FA. Moreover, associated beneficial health effects, such as reduced mouse body weight and improved glucose metabolism, led the authors to conclude that a partial depolymerisation of the wheat aleurone cell wall could be favourable for their metabolism.

These conclusions are consistent with those from Rosa et al. [127], which upon xylanase and feruloyl esterase treatment of aleurone, described a release in FA, both in free and conjugated forms. Furthermore, although SCFA production was not improved, the fast metabolization of FA by the colonic microbiota promoted the production of FA colonic metabolites.

Moreover, these findings are complementary to those from Rosa, et al. [126], whom upon enzymatic treatment of an aleurone-rich fraction, also found an increase in the release of cell-wall bound phenolic acids in conjugated and free forms. However, this was also associated with an increase in the antioxidant capacity of the fraction due to both the released components and the increase in their bioaccessibility.

However, experiments conducted by Vangsøe et al. [133] demonstrated that the enzyme susceptibility to the aleurone cell wall AXs was correlated to its arabinose-to-xylose ratio. More specifically, it seems that xylanase activity was enhanced in the presence of AXs with lower substitution degrees, which are mainly found in the cell walls of the aleurone layer.

Overall, xylanase has mainly been used for the transformation of the aleurone layer in previous experiments. As those enzymes are capable of hydrolysing WUAXs, known for their negative technological effects, this explains their success. Nevertheless, complete hydrolysis should not be performed, as the dough’s stickiness increases, and the crust darkens. Instead, the release of WUAX-hydrolysed fragments of high molecular weight is preferred. By increasing the dough’s viscosity, decreasing the fermentation gas migration rate and thus improving its retention, the dough shows an improved tolerance to fermentation and oven baking. The final products are reported to increase in volume from 15 up to 30% [6].

In conclusion, only a few modification processes were performed on the aleurone layer despite the large range of known physical, chemical and biological processes [18,157]. As the transformation of wheat bran has been extensively studied, it would be beneficial to rely on the obtained results to undertake new experiments focusing on improving the milling layer’s technological and health benefits in bakery products.

However, the experiments were performed on starting materials of different purity, which explains the disparity in the results. It would then be interesting to work on the aleurone-extracting process to make it efficient and reproduceable. Little else can be accomplished on this part besides aiming to erase the traits that are not beneficial to the rheology of the end-products.

## 6. Conclusions

The aleurone layer is a major bran component that exhibits numerous nutritional and potential health benefits. Multiple processes exist and have been patented to extract this layer, but it seems that a choice must be made between the extraction yield and the purity of the fraction. Although its addition to bakery products has been studied and claimed to be beneficial from a nutritional point of view, some technological negative effects still arise, such as a decrease in loaf volume. These are partly due to the presence of a large number of DFs, mainly water-unextractable. However, those side effects could be reduced, and the aleurone-rich flour ingredient’s functionality improved following some transformation processes preceding its addition to a dough matrix.

Despite all the known benefits of the aleurone layer, as proved in the experiments conducted by researchers, only a few applications exist nowadays with even fewer producers and furnishers. This may be related to the challenges of its extraction—more specifically, to the fact that no universal process exists. Even though it may be created, the natural variability of the grain’s constituents and the impact of the environment would still require this process to be adapted to each raw material, as the grain’s technological properties would change in accordance with its composition. In addition, the negative rheological traits observed upon aleurone addition into bakery products may also explain the absence of its use on the market, unless its presence is not communicated as such.

The appeal of the aleurone layer may also be dwindled by its more expensive extraction process when compared with wholegrain flours and other by-milling products. The latter can be nutritionally relevant for the consumer and would be easier to obtain. However, as demonstrated in this review, the health benefits obtained upon aleurone consumption may be higher (i.e., antioxidant capacity, reduced risk of cancer, etc.). In addition, the presence of the outer parts of the grain could increase the concentration of pesticides residues and impair the technological and organoleptic properties of the cereal end-products, thus favouring the use of the innermost part of the bran which shows those undesirable effects to a lesser extent [18].

Perhaps investigating other types of applications, as well as transformations pre- and post-extraction of the milling fraction, could instigate increased interest in this nutritionally beneficial aleurone layer. The lack of an actual market would also make it an opportunity for a company to renew its interest in the aleurone as an ingredient, not unlike bran fibres, provided its rheological properties are improved. Investigating the limits of aleurone layer incorporation to the formulation of other bakery products than bread in a design of experiments format could be interesting.

## Figures and Tables

**Figure 1 foods-11-03552-f001:**
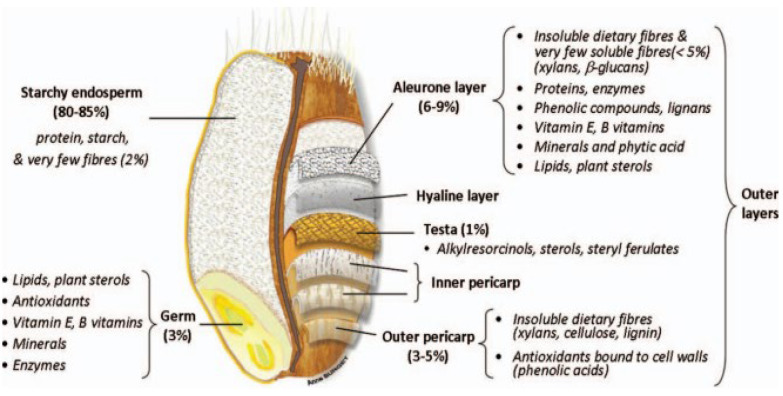
Wheat grain histology [10]. Adapted from Surget and Barron (2005) [21]. (Copyright for reprinting was requested and obtained through Taylor & Francis and Copyright Clearance Center. License number: 5386341293998. License date: 12 September 2022).

**Figure 2 foods-11-03552-f002:**
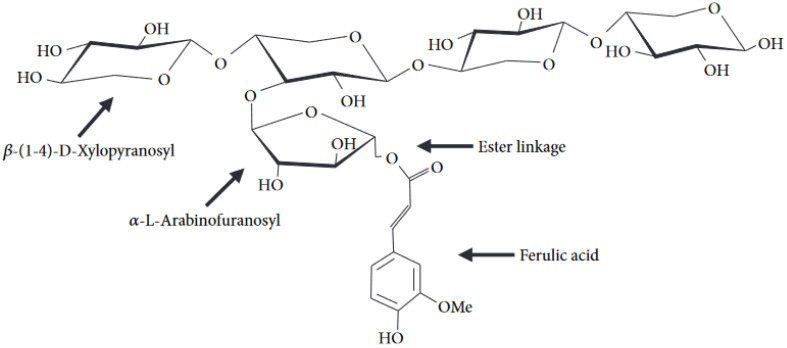
Arabinoxylan structure.

**Table 1 foods-11-03552-t001:** Carbohydrates repartition in the wheat aleurone layer, bran and whole grain.

	Aleurone Layer *		Bran * (%, *w*/*w*)	Whole Grain (%, *w*/*w*)
	Part in Tissue (%, *w*/*w*)	Total in Grain (%, *w*/*w*)		
Arabinoxylan	65 [25,36,37]	-	60–65 [32]	4–9 [36]
β-glucans	29 [36,37]	-	6 [36]	0.5–2.3 [36]
Cellulose	2–3 [32,33,36,37]	≈8 [32,33]	25–30 [32]	2–4 [32]
Glucomannan	2 [36,37]	-	-	<1 [37]
Pentosans	46 [32,33]	≈44 [32,33]	70–80 [32]	8–10 [32]
Starch	0 [32,33]	0 [32]	9–25 [3]	67–71 [32]
Total Dietary Fibre	43–48 [12]	-	40–53 [3]	13 [1]

* The aleurone layer and the bran amount respectively to 5–8% (*w*/*w*) and 5.5–9.5% (*w*/*w*) of the wheat grain [21]. All data were placed in the same unit to facilitate comparison.

**Table 4 foods-11-03552-t004:** Minerals repartition in the wheat aleurone layer, bran and whole grain.

	Aleurone Layer *		Bran * (mg/100 g dm)	Whole Grain (mg/100 g dm)
	Part in Tissue (mg/100 g dm)	Total in Grain (%, *w*/*w*)		
Calcium	73.0 [4,10]	53.3 [4]	100 [32]	30.0–70.0 [32]
Copper	12.4 [4]	67.3 [4]	0.8–1.6 [32,54]	0.4–1.5 [4,32,55]
Iron	19.0–34.0 [4,10,54]	78.6 [4]	5.0–15.0 [3,32,54]	1.8–8.0 [4,32,54]
Magnesium	600 [4,10]	86.7 [4]	500–700 [32]	100–200 [4,32]
Manganese	8.0–13.0 [4,10,54]	53.2 [4]	7.2–14.4 [3,54]	2.4–5.6 [4,54,55]
Sodium	-	-	5.0–30.0 [32]	3.0 [32]
Phosphorus	1400–3170 [4,10,54]	76.2 [4]	900–1500 [1]	218–792 [1]
Potassium	1100 [4,10]	67.6 [4]	1000–1500 [32]	350–600 [32]
Zinc	12.0 [4,10]	68.2 [4]	5.6–50 [3,32,54]	2.1–12 [4,32,54]
Total (%)	12.0 [32,33]	≈44.0 [32,33]	3.39 [1]	1.1–2.5 [1,32]

* The aleurone layer and the bran amount respectively to 5–8% (*w*/*w*) and 5.5–9.5% (*w*/*w*) of the wheat grain [21]. All data were placed in the same unit to facilitate comparison.

**Table 5 foods-11-03552-t005:** Vitamins repartition in the wheat aleurone layer, bran and whole grain.

	Aleurone Layer *		Bran * (mg/100 g dm)	Whole Grain (mg)
	Part in Tissue (mg/100 g dm)	Total in Grain (%, *w*/*w*)		
Thiamin (B1)	1.6 [36,56,65]	32 [3,49,54,66]	0.54 [25]	0.4–0.8 [32]
Riboflavin (B2)	1.0 [36,56,65]	37 [54,66]	0.39–0.75 [25]	0.1–0.2 [32]
Niacin (B3)	61.3–90.2 [36,56,65]	≈80 [3,49,54,66]	14.0–18.0 [25]	4.0–6.0 [32]
Pantothenic acid (B5)	4.51 [36,56]	41 [54,66]	2.2–3.9 [25]	1.0–2.0 [32]
Pyridoxin (B6)	3.6 [36,56,65]	≈60 [3,49,54,66]	1.0–1.3 [25]	0.5–1.0 [32]
Folate (B9)	0.2–0.8 [10]	-	0.079–0.2 [25]	0.035–0.055 [32]
Vitamin E	1.2–2.0 [10]	-	1.4 [25]	2.0–6.0 [32]

* The aleurone layer and the bran amount respectively to 5–8% (*w*/*w*) and 5.5–9.5% (*w*/*w*) of the wheat grain [21]. All data were placed in the same unit to facilitate comparison.

**Table 6 foods-11-03552-t006:** Repartition of bioactive components in the wheat aleurone layer, bran and whole grain.

	Aleurone Layer *		Bran *	Whole Grain
	Part in Tissue	Total in Grain		
Alkylresorcinols (mg/100 g)	≈3.0 [65,67,68]	20–400 [67]	220–400 [1,3]	42–70 [1,50,65]
Phytic acid (g/100 g)	8.4–15.6 [50,65,68]	2.2–5.2 [3]	1.2–1.4 [50,65]	1.18–1.38 [50,65]
Lutein (µg/100 g)	7.2–42.5 [63,68]	-	97–140 [3]	81.9 [63]
Zeaxanthin (µg/100 g)	21.2–77.6 [63,68]	-	25–219 [1]	9.0–43.8 [1,63]
Choline (mg/100 g)	1260 [54]	-	112 [1]	112 [1]

* The aleurone layer and the bran amount respectively to 5–8% (*w*/*w*) and 5.5–9.5% (*w*/*w*) of the wheat grain [21]. All data were placed in the same unit to facilitate comparison.

**Table 8 foods-11-03552-t008:** Overview of existing aleurone-enriched bread formulation processes in literature.

Aleurone-Rich Flour Used	Enrichment Level (%, *w*/*w*)	Basic Flour Type and Protein Content (%)		Bread Formulation Process	Specificities	Reference
Table 7, 1a/1b	20	White flour	11–13	Straight dough	Addition of vital wheat gluten, high fructose corn syrup, and dough conditioner	[12,128]
				Sponge dough	Addition of vital wheat gluten, high fructose corn syrup, ascorbic acid, dough conditioners, mono- and diglycerides	
Table 7, 1a/1b	40 and 75	Wheat flour	11.86	ICC Standard Method 131	-	[13,15]
Table 7, 1a/1b	25 and 50	Wheat flour	12.9	ICC Standard Method 131	-	[16,17]
Table 7, 4	10, 20, 30 and 40	Wheat flour	14.16	-	-	[131]
Table 7, 1a/1b	15, 40, 75 and 100	Conventional bread wheat flour	15.24	Sourdough (MSZ method, 1989)	-	[136]
54.11% purity	18	Wheat flour	-	GB/T 358969-2018 method	Hemicellulase addition (0–60 mg/kg of flour)	[140]

* Basic flour type as described in the corresponding articles, defined as commercial flour.

## Data Availability

Data is contained within the article.

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
