# Peer review of "The Wheat Aleurone Layer: Optimisation of Its Benefits and Application to Bakery Products"

_foods, 2022, doi:10.3390/foods11223552_

Round 1

Reviewer 1 Report

The authors did a great job, their paper is very interesting to read, informative, and backed up with science. I only observed that some of their Tables are misplaced while some are not discussed in the main text. 

Author Response

AUTHOR RESPONSES

We would like to thank the reviewers for taking the time to read our work. We welcome your interesting suggestions for improving this review. Please find below some answers to your comments.

Reviewer 1:

  • L28: does this include frequency of consumption?

The increased consumption of whole grain products may include an enhanced frequency of consumption but it depends on the country’s guidelines (refer to reference [1], p. 78 “The importance of whole-grain cereal product consumption”). This explains the author’s choice of not mentioning it.

  • Tables are misplaced while some are not discussed in the main text.

It seems that a technical error occurred as Table 1 only appeared once in the original paper. Please find enclosed the original layout of the paper in PDF format.

The errors that appeared were automatic references to Tables and Figures. They are now hand-written in order to avoid this issue.

  • Incomplete bibliographic references number 14, 107, 115 and 116.

Thank you for noticing, the patents references were completed.

Reviewer 2 Report

The review summarize the wheat aleurone layer from a nutritional and technological point of view, which will provide information on the elements of understanding to optimize the potential of the wheat aleurone layer .

The current knowledge on the extraction of aleurone layer should also be summarized.

Some of the reference are not shown, L158, 196, 529, 540 etc.

In the conclusion part, the author showed suggest more about the future studies.

Author Response

AUTHOR RESPONSES

We would like to thank the reviewers for taking the time to read our work. We welcome your interesting suggestions for improving this review. Please find below some answers to your comments.

Reviewer 2:

  • The current knowledge on the extraction of aleurone layer should also be summarized.

This topic is discussed in the part 4.1 Extraction of the Aleurone Layer and Its Challenges, from L400 to L496 of the corrected paper.

  • Some of the reference are not shown, L158, 196, 529, 540 etc.

The errors that appeared were automatic references to Tables and Figures. They are now hand-written in order to avoid this issue.

  • In the conclusion part, the author showed suggest more about the future studies.

As per the reviewer’s demand, additional suggestions were added on the last concluding paragraph, L804-810.

Reviewer 3 Report

MS STRENGTHS:

Well written, detailed, comprehensive summary of findings, extensive literature review, critical approach.

MS WEAKNESSES:

- Technical errors: Table 1 repeated several times (highlighted in red, see the file attached), occurrence of Error reports instead citation numbers in the text.

- Sections 4.2.2. and 4.2.3.: It would be useful to indicate the levels of aleurone-rich flour incorporated in dough/bread when discussing dough behaviour /bread characteristics. An overview of used enrichment levels, type of aleurone-rich flour,  basic flour type i.e. more details on the reported formulations of dough/bread, organized in tabulated form would be very informative for the readers.

- Section 5. Optimization of the aleurone layer's potential, Lines 715-733. When mentioning that milling/micromilling, grinding (ultra fine, super fine) are used as treatments to particle size reduction of aleurone fraction, it would be useful to mention the difference btw these techniques, final particle size ranges, etc.

It would be also useful to make more comparisons with wholegrain dough/bread to conclude whether it is necessary to make such complicated extractions of anatomic parts of wheat kernel or would it be more practical to use minimally processed native, wholegrain grains/flours, with preserved natural structure as much as possible.

Some minor corrections are highlighted in the attached file.

Author Response

We would like to thank the reviewers for taking the time to read our work. We welcome your interesting suggestions for improving this review. Please find below some answers to your comments.

Reviewer 3:

  • Technical errors: Table 1 repeated several times (highlighted in red, see the file attached), occurrence of Error reports instead citation numbers in the text.

It seems that a technical error occurred as Table 1 only appeared once in the original paper. Please find enclosed the original layout of the paper in PDF format.

The errors that appeared were automatic references to Tables and Figures. They are now hand-written in order to avoid this issue.

  • Sections 4.2.2. and 4.2.3.: It would be useful to indicate the levels of aleurone-rich flour incorporated in dough/bread when discussing dough behavior/bread characteristics. An overview of used enrichment levels, type of aleurone-rich flour, basic flour type i.e. more details on the reported formulations of dough/bread, organized in tabulated form would be very informative for the readers.

Thank you for this interesting input. An additional Table was created (Table 8) following your recommendations, with the available information in the literature.

  • Section 5. Optimization of the aleurone layer's potential, Lines 715-733. When mentioning that milling/micromilling, grinding (ultrafine, super fine) are used as treatments to particle size reduction of aleurone fraction, it would be useful to mention the difference btw these techniques, final particle size ranges, etc.

We understand your comment on this topic, however it seemed more relevant to only mention that these techniques have the same aim of obtaining a certain particle size rather than explaining the intricacies of each. Indeed, as each technique can be used to obtain large ranges of particle sizes, the aim of each study then defines the parameters used. The paragraph was rewritten to ease understanding.

  • It would be also useful to make more comparisons with wholegrain dough/bread to conclude whether it is necessary to make such complicated extractions of anatomic parts of wheat kernel or would it be more practical to use minimally processed native, wholegrain grains/flours, with preserved natural structure as much as possible.

Thank you for bringing this notion to our attention. The Tables describing the composition of the aleurone layer partly respond to this, as it shows the nutritional relevance of the aleurone layer compared to the whole grain or the bran. However, the notion of wholegrain bread is not homogenous among studies and the experiments conducted by researchers use different breadmaking methods, thus creating a bias if used for comparison. The diversity of aleurone-enriched flour purity used for the breads and the breadmaking process itself also add to the complexity of comparison. Nonetheless, this consideration was added to the review, on L602-604.

Reviewer 4 Report

The paper is well-written and covers an interesting subject. However, there are some technical problems, which cause a number of references to disappear from the current version of the manuscript. The non-functional references are placed on lines: 158, 196, 228, 229, 233, 360, 385, 399, 528, 540,

Additionally, I would recommend adding references to the recent reviews related to the internal structure of wheat grain:

https://doi.org/10.1016/j.jcs.2019.102869

arabinoxylans:

https://doi.org/10.1080/10408398.2018.1555134

https://doi.org/10.1021/acs.jafc.1c04506

as well as the new methods of whole grain pre-treatment:

https://doi.org/10.1111/1541-4337.12625

including microfluidization

https://doi.org/10.1080/10408398.2018.1555134

It would also be good to indicate recent developments in colored wheat, which include varieties with blue aleurone layer:

https://doi.org/10.3389/fnut.2022.878221

https://doi.org/10.1080/10408398.2020.1793727

https://doi.org/10.1016/s1673-8527(08)60149-6

You could also mention that the aleurone layer may be regarded as an important source of high-quality protein in bran-derived isolates, such as described in https://doi.org/10.1016/j.foodchem.2019.03.020 (high nutritional quality of aleurone proteins has already been referred).

Author Response

AUTHOR RESPONSES

We would like to thank the reviewers for taking the time to read our work. We welcome your interesting suggestions for improving this review. Please find below some answers to your comments.

Reviewer 4:

  • There are some technical problems, which cause a number of references to disappear from the current version of the manuscript. The non-functional references are placed on lines: 158, 196, 228, 229, 233, 360, 385, 399, 528, 540.

The errors that appeared were automatic references to Tables and Figures. They are now hand-written in order to avoid this issue.

  • I would recommend adding references to the recent reviews related to the internal structure of wheat grain: https://doi.org/10.1016/j.jcs.2019.102869

Arabinoxylans:

https://doi.org/10.1080/10408398.2018.1555134

https://doi.org/10.1021/acs.jafc.1c04506

New methods of whole grain pre-treatment: https://doi.org/10.1111/1541-4337.12625

Microfluidization: https://doi.org/10.1080/10408398.2018.1555134

Recent developments in colored wheat, which include varieties with blue aleurone layer: https://doi.org/10.3389/fnut.2022.878221

https://doi.org/10.1080/10408398.2020.1793727

https://doi.org/10.1016/s1673-8527(08)60149-6

The aleurone layer may be regarded as an important source of high-quality protein in bran-derived isolates, such as described in https://doi.org/10.1016/j.foodchem.2019.03.020 (high nutritional quality of aleurone proteins has already been referred).

Thank you for this relevant additional literature, it has been added to the review.

Round 2

Reviewer 4 Report

I find the current version of the manuscript appropriate for publication.

Best regards

Author Response

The addition of these references was reviewed in the text. However, as we are not able to get full access to one of the articles (https://doi.org/10.1021/acs.jafc.1c04506), it has thus not been added to the review.